# A Unified Equation for Predicting Crack Growth in Rubber Composites Across All Crack Growth Rates

**DOI:** 10.3390/polym17101357

**Published:** 2025-05-15

**Authors:** Aaron M. Duncan, Keizo Akutagawa, Dimitrios G. Papageorgiou, Julien L. Ramier, James J. C. Busfield

**Affiliations:** 1School of Engineering and Material Science, Queen Mary University of London, Mile End Road, London E1 4NS, UK; k.akutagawa@qmul.ac.uk (K.A.); d.papageorgiou@qmul.ac.uk (D.G.P.); 2SLB Cambridge Research, Cambridge CB3 0EL, UK; jramier@slb.com

**Keywords:** crack growth rate, tearing energy, energy dissipation, failure prediction

## Abstract

The relationship between tearing energy and crack growth rates in elastomers is typically divided into three regions—slow crack growth, fast crack growth, and a transitional region—each described by separate power law relationships, requiring six variables to fully characterize the behavior. This study introduces a novel, unified equation that simplifies this relationship by combining two coexisting energy dissipation mechanisms into a single model with only four variables. The model consists of two terms, one for each energy dissipation mechanism: one term is dominant at slow crack growth rates and limited by a threshold energy, and the other is dominant at fast speeds. The transition region emerges naturally as the dominant mechanism shifts. The model’s simplicity enables new advances, such as predicting fast crack growth tearing and transition energies using only slow crack growth data. This capability is demonstrated across a wide range of non-strain crystallizing rubbers, including filled and unfilled compounds, tested at room temperature and elevated temperatures and in both swollen and unswollen states. This model offers a practical tool for material design, failure prediction, and reducing experimental effort in characterizing elastomer performance. Notably, this is the first model to unify slow, transition, and fast crack growth regimes into a single continuous equation requiring only four variables, enabling the prediction of high-speed behavior using only low-speed experimental data—a major advantage over existing six-parameter models.

## 1. Introduction

### 1.1. Tearing Energy Versus Crack Growth Rate

The relationship between the crack growth rate and tearing energy of elastomers has been extensively studied, and it has been shown that there exists a power law relationship between tearing energy and crack growth rate [1,2,3,4,5,6,7]. This relationship is shown in Equations (1) and (2):(1)$$\dot{c}=B{\left(T\right)}^{\beta }$$(2)$$T={\left(\frac{\dot{c}}{B}\right)}^{\frac{1}{\beta }}$$
where T and $\dot{c}$ are the tearing energy and crack growth rate, respectively, and B and β are the material parameters. Higher values for B represent higher tearing energies at all crack growth rates. Higher values of β indicate that crack growth rate has a larger impact on tearing energy.

This power law relationship has been shown to effectively predict the crack growth rate as a function of the tearing energy within specific crack growth rate limits in a wide range of studies in the literature, including for both unfilled and filled rubber compounds [1,2,8] in both the swollen and unswollen state [9] and at a wide range of temperatures [8,10,11,12]. Equation (3), derived from Equation (2), shows that the power law relationship between T and $\dot{c}$ produces a linear fit on a logarithmic plot of T and $\dot{c}$. The gradient of this linear fit is equal to 1/β, and the intercept is equal to $-1/\beta [log\left(B\right)\;]$.(3)$$\log\left(T\right)=\frac{1}{\beta }\left[\log\left(\dot{c}\right)-\log\left(B\right)\right]$$

Across a wide enough range of crack growth rates, non-strain crystallizing rubbers have been reported to show up to three distinct regions: a slow crack growth region at low tearing energies, followed by a transition region in which the crack growth rate tends to increase dramatically with small increases in tearing energy, before entering a fast crack growth region at higher tearing energies [1,3,13,14,15].

Crack growth can follow one of four typical behaviors, as shown in Figure 1.

During slow crack growth, the crack often exhibits a steady but rough crack path (see Figure 1b), typically caused by crack path undulations or microvoid formation near the crack tip. In the transition region, the crack can continue to display steady crack growth for some rubbers (see Figure 1b) but often displays stick slip crack growth; see Figure 1c. Stick slip is characterized by alternating periods of rapid and slow crack growth. This instability arises from local crack tip toughening mechanisms—such as strain-induced crystallization or energy dissipation in the process zone—that temporarily halt crack advancement until sufficient energy accumulates to overcome resistance. The result is a repeating cycle of crack arrest and sudden re-initiation, producing a saw-tooth pattern in tearing energy versus time. Fast crack growth is usually characterized by a steady smooth crack surface (see Figure 1a), as the crack advances stably with minimal deflection. At high tearing energies, the crack tip experiences sufficient driving force to propagate continuously without interruption, limiting time for microstructural rearrangements or void formation ahead of the tip. This results in a straight crack path and smooth fracture face. However, it is noted that under some material compositions or loading conditions, fast crack growth can exhibit more complex or rough fracture [3,13,17,18]. In some extreme cases, the rubber can exhibit an extreme version of stick slip, where the crack deviation is significantly larger, resulting in knotty tearing; see Figure 1d. None of the data points presented in this paper are the result of knotty tearing.

It is widely reported that slow crack growth and fast crack growth have their own separate mechanisms that govern the variables used in Equations (1) and (2) and that, in the transition region, the crack flips repeatedly and rapidly between the two mechanisms [3,14,15,19]. Note that some rubbers have been reported in the literature to only show a fast and slow region, with no transition between the two [13].

Figure 2 shows all three regions for an unfilled styrene-butadiene rubber (SBR). All regions are treated as separate regions, each with their own two variables to define the gradient and intercept on the logarithmic axes. Fully defining all three regions requires six variables; see Equation (4). The determination of the threshold energies that define the transition point between regions ($T_{ds}\;$ and $T_{df}$) affects the fit of the power law relationship and, therefore, the calculation of the six variables. This can make studying the relationship between tearing energy and crack growth rate over a wide range of crack growth rates, or near the transition energies, challenging. To fully define the crack growth behavior over all three regions, Equation (4) must be used:(4)T=c˙Bs1βs                if T≤Tdsc˙Bt1βt     if Tds≤T≥Tdfc˙Bf1βf                 if T≥Tdf    
where $B_{s}$ and $\beta_{s}$ are material parameters and are determined by the slow tearing behavior, $B_{t}$ and $\beta_{t}$ are material parameters and are determined by the transition tearing behavior, and $B_{f}$ and $\beta_{f}$ are material parameters and are determined by the fast tearing behavior.

Accurately predicting the threshold energy at transition point $T_{ds}$ is very important. As previously mentioned, the crack growth rate rises rapidly in this region, often increasing by one to three orders of magnitude, with just a 5 to 25% increase in tearing energy for many rubbers [1,13,14,15]. Currently, the only way to establish this energy is to test, at multiple speeds, in both the slow region and transition region and to plot both on logarithmic axes to obtain the transition point. Since the speed of the threshold energy is not known ahead of time, this often requires designing testing plans that test over many orders of magnitude of crack growth rates to ensure $T_{ds}$ is identified.

The novelty of this work lies in the development of a single, simplified equation that models the full spectrum of crack growth behaviors—slow, transitional, and fast—using only four variables.

### 1.2. Viscoelastic Energy Dissipation During Crack Growth

The tearing energy of viscoelastic materials results from both the energy required to create new crack surfaces and from irreversible energy loss, known as hysteresis, when rubber either side of the crack is unloaded during crack propagation. The minimum energy (sometimes referred to as the threshold tearing energy) needed to create new crack surfaces is approximately 50 J/m^2^ for a wide range of rubbers [20], and in most cases, it is the smaller of the two contributions to the total tearing energy.

Dynamic mechanical analysis can be used to demonstrate the relationship between the irreversible energy (loss modulus), and the loading and unloading frequency of rubber samples. All rubbers show a similar trend between loss modulus and frequency: at higher frequences, the loss modulus increases by orders of magnitude as frequency increases, before reaching a small local maximum and then reaching a plateau region where the loss modulus does not increase further with higher frequencies [21].

At room temperature, the loss modulus plateau values are not reached until very large frequencies, equivalent to strain rates in the order of 1000% strain per second. Since these higher strain rates produce significantly more energy loss than at the slower strain rates, and the highest strain rates during crack growth are found near the crack tip, it has been proposed by Rivlin and Thomas [7] that irreversible energy changes due to crack propagation occur most significantly near the crack tip. This can explain why measurement of tearing energy versus crack growth rate are independent of the test piece geometry. Since only the rubber near the crack tip contributes significantly to the total tearing energy, the shape of the sample further from the crack tip does not impact the tearing energy results.

### 1.3. Tearing Energy Experiment Geometries

As stated in Section 1.2, due to the energy dissipation during crack growth occurring almost exclusively near the crack tip, the shape of the sample does not affect the measured tearing energy of rubber samples. As a result, different geometries can be chosen to measure tearing energy, with each geometry having its own advantages and disadvantages. Rivlin and Thomas [7] further showed that different crack geometries and loading configurations can be considered equivalent to a local opening (Mode I) fracture at the crack tip, allowing tearing energy to be treated as a material property independent of test mode. This paper will utilize the trouser tear (TT) test for measuring tearing energy and will use data from literature collected with pure shear crack growth (PSCG) tests. Figure 3 shows the shape of these samples.

No direct experimental comparison between PSCG and TT tests was conducted within this study; however, their equivalence in measuring tearing energy is well established in the literature [7,16], ensuring the validity of combining data from both methods.

A detailed description of the samples and how tearing energy is derived for each is given by Rivlin and Thomas [7]. A fundamental distinction between the two experimental designs is the assignment of independent (controlled) and dependent (measured) variables. During PSCG tests, the tearing energy is controlled by the extension between the grips and stays constant throughout the experiment. The crack growth rate is then measured by recording the experiment using a video camera. During TT tests, the crack growth rate is controlled by setting the speed of separation of the grips, which typically remains as a constant during the experiment. The tearing energy is determined by measuring the force required to separate the grips. PSCG tests effectively control the tearing energy, whilst TT tests control the crack growth rate. TT tests require less material and are easier to conduct as they do not require a high-speed camera, which can complicate the experimental set-up, particularly at elevated temperatures in an oven. However, TT tests are limited in crack growth rate by the maximum speed of the cross-head displacement, whereas high-speed cameras allow PSCG tests to achieve very high crack growth rates.

Many studies that examine the effect of crack growth rate on tearing energy utilize PSCG tests to achieve the highest crack growth rates. However, in the transition region, small changes in crack growth rate can create a large change in the tearing energy, making it challenging to collect data in this region and hence to accurately determine the transition energy and speed. In the literature, TT tests are less commonly used to investigate wide ranges of crack growth rate. Due to the easier set-up and the ability to easily explore the transition region, TT tests were chosen for this study.

## 2. Materials and Methods

### 2.1. New Method for Modeling Tearing Energy Versus Crack Growth Rate

In this work, a new model is proposed to describe the relationship between crack growth rate and tearing energy across all three crack growth regions, using fewer variables and transition energies than the previous model. In the new model, Equation (6) is used to model all crack growth regions. The equation comprises two terms, each contributing to the total tearing energy. The first term ($T_{s}$) and second term ($T_{f}$) represent two energy loss mechanisms that dominate at slow and fast crack growth rates, respectively. Unlike the current methods, both slow and fast crack growth terms are active across all crack growth rates. This indicates that rather than distinct transitions in mechanisms, both mechanisms coexist, with one being more dominant depending on the crack growth rate.(5)$$T=T_{s}+T_{f}$$

$T_{s}$ and $T_{f}$ are defined by Equations (6) and (7), respectively:(6)$$T_{s}={\left(\frac{\dot{c}}{B_{s}}\;\right)}^{1/\beta_{s}}\;\mathrm{if}\;T_{s}\leq T_{d}\mathrm{else}\;T_{s}=T_{d}$$(7)$$T_{f}={\left(\frac{\dot{c}}{B_{f}}\;\right)}^{1/\beta_{f}}$$
where $B_{f}$, $\beta_{f}$, $B_{s}$ and $\beta_{s}$ are material parameters, and $T_{d}$ is the threshold tearing energy.

While the model retains a threshold energy $T_{d}$ for the slow crack growth term, it eliminates the need to define a separate transition point for fast crack growth (i.e., $T_{df}$ in previous models). Additionally, as will be discussed in Section 3.2, $T_{d}$ can be predicted from slow crack growth data, further reducing the experimental complexity. The threshold tearing energy $T_{d}$ represents the upper limit of the slow crack growth energy dissipation mechanism. The contribution to the total tearing energy from the slow crack growth energy dissipation mechanism cannot increase beyond this energy, even at higher crack growth rates.

Figure 4 presents this model on logarithmic axes, utilizing the data from Figure 2. It displays both terms separately with dashed lines and also shows their sum. At slow crack growth rates, $T_{s}\gg T_{f}$; therefore, T can be approximated as $T_{s}$. At high crack growth rates, $T_{s}\ll T_{f}$; therefore, T can be approximated as $T_{f}$. Therefore, these two regions appear as straight lines when plotted using logarithmic axes. The transition region, however, does not appear as a straight line, but rather as a curved line with two asymptotes: one at $\log\left(T\right)=\log\left(T_{d}\right)$ and the other at $\log\left(T\right)=\frac{1}{\beta_{f}}\left[\log\left(\dot{c}\right)-\log\left(B_{f}\right)\;\right]$.

A key point to highlight is that the gradient of the asymptote in Figure 4 is greater than that used to model fast crack growth in Figure 2, despite the use of the same data. This is because the data points between 10^2^ and 10^3^ mm/s, labeled as “non-asymptotic data points”, have been included in the fast crack growth region in Figure 2, but in Figure 4, we can see that these points have not yet converged closely to the asymptote. This will result in different calculated variables for $\beta_{f}$ and $B_{f}$. This paper shows that the values calculated with this method can be fixed to a value of 4/3 for $\beta_{f}$ for all the rubbers examined in this paper. Fixing this parameter reduces the number of variables needed to fit per material down to just four. The value of $\beta_{f}=4/3$ was selected empirically, as it consistently provided good fits across all materials studied. While this exponent aligns broadly with expectations for viscoelastic fracture behavior, a detailed theoretical derivation for this specific value remains an open question.

Figure 5 shows the same data and fits as Figure 4 but in a semi-log plot, and this helps to visualize the magnitude of the contributions of each term to the total tearing energy.

### 2.2. Materials

Table 1 provides a description of the materials used in this study. Data for materials marked with a “*” were digitized from plots taken from Tsunoda [14], and more information on those materials can be found in Tsunoda [14]. Data marked with a “†” were digitized from plots taken from Kadir and Thomas [13], and more information on the nitrile butadiene rubber (NBR), styrene-butadiene rubber (SBR), and butadiene rubber (BR) used can be found in the work of Kadir and Thomas [13]. All other NBR materials were prepared by mixing uncured NBR with 33% acrylonitrile content and the typical small amounts of plasticizer, activators, accelerators, and antioxidants that would be widely adopted in commercial practice in an internal mixer, before they were briefly masticated using a two-roll mill prior to being cured into 2 mm thick sheets using a hot press. The SBR samples named SBReV and SBRcV, which contain different curing systems to produce a soft and hard compound, were likewise prepared by mixing uncured SBR in an internal mixer and briefly masticated using a two-roll mill prior to being cured into 2 mm thick sheets using a hot press.

### 2.3. Trouser Tear Tests

The NBR, SBReV, and SBRcV data presented in this study were generated through original TT tests, while additional datasets were digitized from published literature sources, as noted in the figure captions and text. The TT tests were conducted in accordance with ASTM D1938, (2019) [22] at crack growth rates ranging from 0.01 mm/s to 10 mm/s. The tearing energy values were calculated by considering the propagation energy as defined by Windslow et al. [23]. The tearing energy data for the other materials marked with a “*” or “†” were collected by digitizing plots from Tsunoda [14] and Kadir and Thomas [13], respectively, both of whom used PSCG tests to collect tearing energy data. To maximize the mapping of the crack growth rate versus tearing energy plot, one test was conducted at each crack growth rate at every Log(0.25 mm/s) interval, following common practice for this type of study. Samples showing significant early crack growth deviation, for which accurate identification of the propagation energy was difficult, were removed from the dataset. For digitized literature data, reproducibility follows that reported by the original authors.

## 3. Results

### 3.1. Fitting of Tearing Energy Data

Figure 6, Figure 7 and Figure 8 show tearing energy versus crack growth data from the materials in Table 1, with fits made using Equation (5). Since, at slow tearing, $T_{s}\gg T_{f}$, values for $B_{s}$ and $\beta_{s}$ can be found by applying a linear fit to the slow tearing region and taking the gradient and intercept to fit $\beta_{s}$ and $-1/\beta_{s}\;[\log\left(B_{s}\right)],$ respectively, while ignoring the values of $T_{d}$, $B_{f}$, and $\beta_{f}$. Once values for $B_{s}$ and $\beta_{s}$ have been established, the other variables can be fit. Good fits for all material in this paper could be obtained by setting $\beta_{f}$ to a value of 4/3, so this has been applied for all fittings.

Note that, in Figure 6, the BR_AT material has no transition region; the same conclusion was drawn by Kadir and Thomas [13]. This can be modeled using Equation (5) if the threshold energy $T_{d}$ is high enough that the fast tearing energy term $T_{f}$ is dominant over the slow tearing energy term $T_{s}$ before reaching the threshold term.

Figure 6 and Figure 7 demonstrate that, despite employing fewer variables than Equation (4) and featuring one instead of two transition points, Equation (5) effectively captures the tearing energy versus crack growth behavior across a broad range of crack growth rates. Figure 6 illustrates this capability for NBR, SBR, and BR, even in the case of BR, which exhibits no transition region. Figure 7 shows that Equation (6) provides good fits for SBR, both with and without fillers, across a range of temperatures and for both swollen and unswollen samples.

Material SBR0.4vr, shown in Figure 7d, shows no obvious transition region or distinction between fast and slow tearing energy, and therefore, no attempt to fit the transition energy or fast tearing energy terms is made. This material is revisited in Section 3.3.

Figure 8 shows tearing energy data from TT tests. As stated in Section 1.3., TT tests are not able to reach speeds as high as those in PSCG tests. As a result, the data collected with this method are insufficient for validating the fast crack growth region, which represents a limitation of this study that must be addressed in future work. These data can still be useful for exploring the link between slow tearing energies and the transition energy $T_{d}$. For all the fittings in Figure 8, the fast tearing energy term $T_{f}$ is set to zero.

Table 2 shows the values used in the fits of Figure 6, Figure 7 and Figure 8.

### 3.2. Relating Variables

Employing Equation (6) to model tearing energy versus crack growth rate offers the advantage of fewer variables and only two distinct crack growth regions. This simplified model raises the possibility of investigating a potential link between these two regions and, consequently, whether it is feasible to predict the behavior of one region from the other. This investigation can be conducted by examining the correlation between the variables used to fit the slow and fast tearing energy regions. Figure 9 and Figure 10 explore this relationship.

Figure 9 shows the values for $\log\left(B_{S}\right)$ and $\log\left(B_{f}\right)$ used in the fits of Figure 7. As Figure 9 shows, there is an approximately linear relationship between $\log\left(B_{s}\right)$ and $\log\left(B_{f}\right)$.

Figure 10 shows the values used for $\log\left(B_{s}\right)$ plotted against the crack growth rate at which the transition energy is reached $\left({\dot{c}}_{Td}\right)$ for the materials in Figure 7 and Figure 8. The linear fit shown in Figure 10 is found for literature data plotted in Figure 7 only, but as can be seen the data from Figure 8, materials also fit to the same linear relationship. The slow tearing energy data for Figure 6 had too little data to reliably fit slow tearing energy data accurately and, therefore, have not been included in this analysis.

As Figure 9 and Figure 10 show, there is a linear relationship between $\log\left(B_{s}\right)$ and both $\log\left(B_{f}\right)$ and $\log\left({\dot{c}}_{Td}\right)$. This means that $\log\left(B_{f}\right)$ and $\log\left({\dot{c}}_{Td}\right)$ can be calculated using Equations (8) and (9). Given ${\dot{c}}_{Td}$, $B_{s}$, and $\beta_{s}$ the value for $T_{d}$ can be calculated using Equation (10); if the linear relationships shown are applied, then all values used in Equation (5) can be predicted from using just $B_{s}$ and $\beta_{s}$, which themselves can be found from just the slow tearing energy data. Alternatively, it is also possible to work backwards: if the values of $B_{f}$ and $T_{d}$ are known, predictions for $B_{s}$ and $\beta_{s}$ can also be calculated.(8)$$\log\left(B_{f}\right)=0.252\log(B_{s})+3.288$$(9)$$\log\left({\dot{c}}_{Td}\right)=0.295\log(B_{s})+0.428$$(10)$$\log\left(T_{d}\right)=\frac{1}{\beta }\left[\log\left({\dot{c}}_{Td}\right)-\log\left(B\right)\right]$$

### 3.3. Predicting Transition and Fast Crack Growth Tearing Energies from Slow Tearing Energy

Figure 11 and Figure 12 replot the data shown in Figure 7 and Figure 8. The figures also now show fits using Equations (8)–(10) to identify the variables in Equation (5). The values for $B_{s}$ and $\beta_{s}$ were again set by using a linear fit on the slow tearing region and taking the gradient and intercept to fit $\beta_{s}$ and $\log\left(B_{S}\right)$, respectively. The values for $B_{f}$ and $T_{d}$ were calculated using the linear relationships derived from Figure 11 and Figure 12. The value for $\beta_{f}$ was set to 4/3 for all materials.

Figure 11 shows that this method produces good fits at faster tearing energies, with sensible predictions for transition energies, as well as reasonable fits at higher crack growth rates. It is noted in Section 3.1 that no obvious transition region or shift to fast tearing energy region is identifiable for SBR0.4vr. Figure 11d shows that, despite this, the method used to fit transition energy and fast tearing energies still produces sensible predictions.

Although the data used in Figure 12 were not fast enough to evaluate the higher tearing energy predictions, it can be seen that the fitting method produced sensible predictions for the transition tearing energy, and the experimental data were mapped well over the measured range.

## 4. Discussion

The value of 4/3 for $\beta_{f}$ and the relationship between $\log\left(B_{S}\right)$ to $\log\left(B_{f}\right)$ and $\log\left({\dot{c}}_{Td}\right)$ are important findings and could have a significant impact on predicting fast tearing energy and transition energies. Of particular use is the ability to predict the transition point at the end of slow crack growth. Without sufficient data at both speeds above and below the transition point, determination of the transition point can be difficult. As the transition point is not known ahead of testing, it can be difficult to ensure that these data will be collected. In the example of NBRN550(125C), shown in Figure 8a, the transition point was at a high enough crack growth rate that it was approaching the upper limit of the testing facilities. A more reliable method for predicting the transition point may be to collect data at slow tearing energy speeds and then use the method developed in this paper to predict the transition speed and energy. However, fast constant tearing energy experiments, especially those including slow tearing energy data, are not common in the literature, and more testing on a wider range of elastomers is needed to confirm that this approach works for all materials.

The model uses a fixed value of $\beta_{f}=4$/3 for the fast crack growth exponent, which provided consistently good fits across all tested rubbers, as shown in Table 2. This choice simplifies fitting by reducing the number of free parameters. However, while the value aligns well with the viscoelastic fracture behavior of non-strain crystallizing rubbers, different material systems with unique fracture mechanisms, such as strain induced crystallization, may require re-evaluation of this exponent. Further testing across a broader set of elastomers and fracture modes will help confirm the universality of this fixed exponent.

While the effect of crack growth rate on tearing energy is well understood, the precise effect of crack growth rate on the underlying mechanisms is still under discussion. Andrews [24] gives an in-depth theory on tearing energy for viscoelastic materials and shows that the higher tearing energies seen for rubber materials result from irreversible energy losses during the loading and subsequent unloading of rubber during crack growth. Andrews also put forward the idea that higher crack growth rates result in higher strain rates, which increases energy dissipation losses. Efforts have been made to explain how different crack growth regions arise from these mechanisms [25,26]; however, more research into this is needed. Equation (5) implies that the total tearing energy has two mechanisms that are both present at all crack growth rates, rather than, as Equation (4) would suggest, multiple mechanisms that are active or inactive at different crack growth rates. A better understanding of how the crack growth rates alter the strain rates not only at the crack tip but also in the body of the sample removed from the crack tip, alongside a better understanding of the subsequent effect of this on any dissipative losses that toughen the materials, is still needed. This remains a point of focus in our research activity, which will be more fully explored using a finite element approach in a subsequent paper.

## 5. Conclusions

This paper presents a new equation for modeling crack growth rate versus tearing energy. Existing models create three distinct regions of crack growth: slow crack growth, fast crack growth, and a transitional region. Existing models describe the relationship using a simple power law for each region, resulting in six variables to characterize all crack growth rates. This approach treats each region separately, necessitating the determination of transition points, which complicates the analysis. This work introduces a novel approach that simplifies the modeling of all crack growth rates using only four variables. The equation comprises two terms, each contributing to the total tearing energy. The first term dominates at slower crack growth rates and is limited by a threshold energy; the second term dominates at faster crack growth rates. The transition region appears after the threshold and before the fast crack growth dominates the resulting behavior.

The new equation is shown to provide good fits for a wide range of non-strain crystallizing rubbers, including styrene-butadiene rubber (SBR) with and without fillers at different temperatures, unfilled SBR in different swollen states, SBR with different crosslinking systems, and unfilled butadiene rubber and nitrile butadiene rubber in both swollen and unswollen conditions at different temperatures and with different carbon black fillers. This broad applicability demonstrates the universality of the approach, making it a valuable tool for predicting crack growth behavior across diverse rubber formulations and processing conditions.

Unlike current methods, both slow and fast crack growth terms are active across all crack growth rates. This indicates that rather than there being distinct transitions in the mechanisms, both mechanisms coexist, with one dominating depending on the crack growth rate.

The gradient or $\beta_{f}$ term for fast crack growth has a wide range of values reported in the literature, with value being, in part, dependent on the choice of the transition energy $T_{df}$ used to define the end of the transition region and beginning of fast crack growth. It is not necessary to define this point in the model used here, and as such, the values calculated with this method are more consistent across a wide range of non-strain crystallizing rubbers. All of the rubbers analyzed in this paper were found to have good fits with a value of 4/3 for $\beta_{f}$. Fixing this parameter reduces the number of variables needed to fit each material down to just four.

The parameters used to fit slow crack growth data have been found to correlate to the transition energy and parameters used to fit fast tearing energy data. This makes it possible to predict fast tearing energy behavior from slow tearing energy data; this paper shows these predictions have close fits to the data for all the different rubbers analyzed.

## Figures and Tables

**Figure 1 polymers-17-01357-f001:**
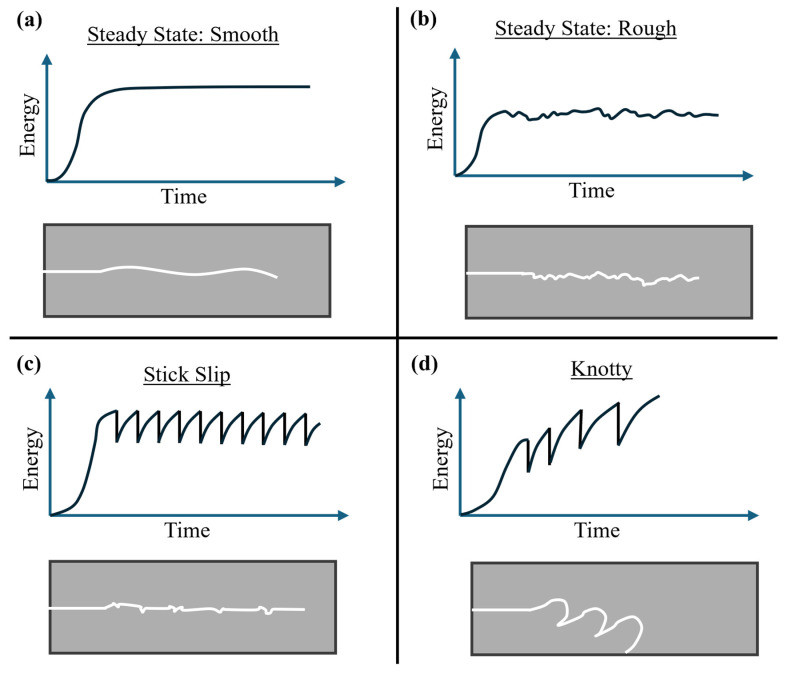
Examples of crack growth behavior showing time versus energy plots, and crack growth paths after an initial straight flaw, under the energy versus time graphs. The figure shows examples of (**a**) smooth steady state tearing, (**b**) rough steady state tearing, (**c**) stick slip tearing, and (**d**) knotty tearing. Figure adapted from Sakulkaew [16].

**Figure 2 polymers-17-01357-f002:**
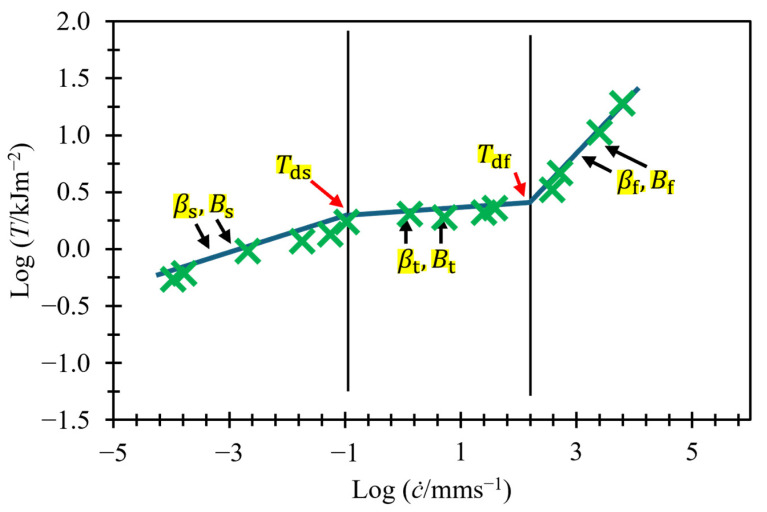
Crack growth rate versus tearing energy for unfilled styrene-butadiene rubber (SBR) (data taken from Tsunoda [14]).

**Figure 3 polymers-17-01357-f003:**
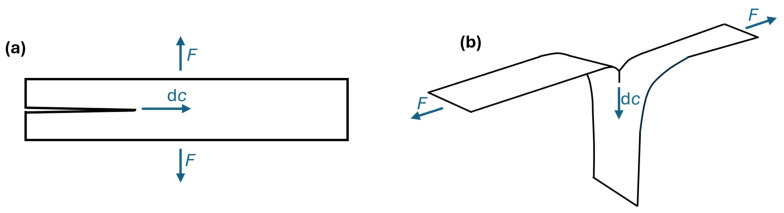
Diagram to show sample shape, direction of force (*F*), and direction of crack growth (d*c*) for our (**a**) pure shear crack growth experiment and (**b**) trouser tear tests.

**Figure 4 polymers-17-01357-f004:**
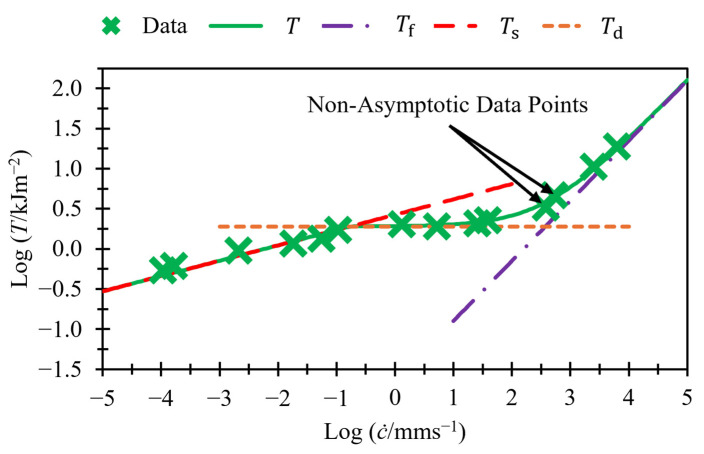
Crack growth rate versus tearing energy for unfilled SBR (data taken from Tsunoda [14]). Fits made using Equation (5).

**Figure 5 polymers-17-01357-f005:**
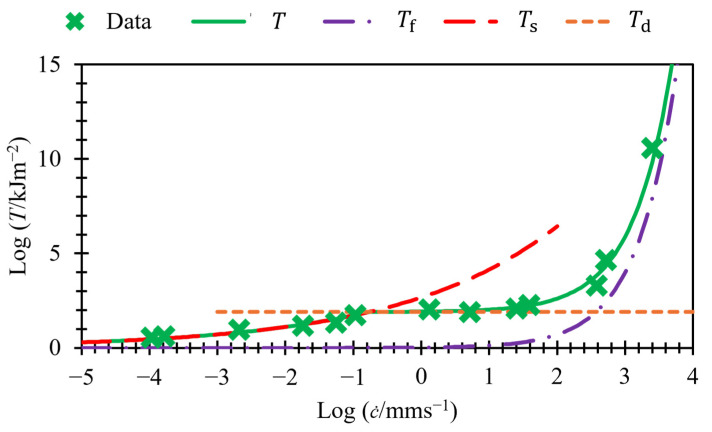
Semi-log plot of crack growth rate versus tearing energy for unfilled SBR (data taken from Tsunoda [14]), showing the individual components of Equation (5).

**Figure 6 polymers-17-01357-f006:**
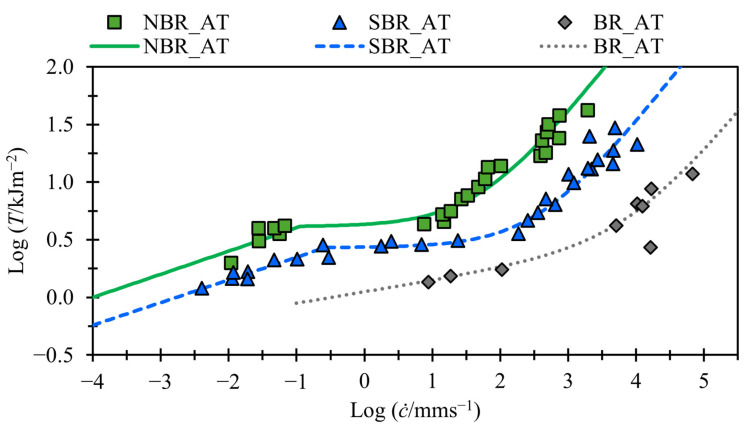
Tearing energy vs. crack growth rate for unfilled NBR, SBR, and BR compounds (data taken from Kadir and Thomas [13]). Fits shown with solid and dashed lines.

**Figure 7 polymers-17-01357-f007:**
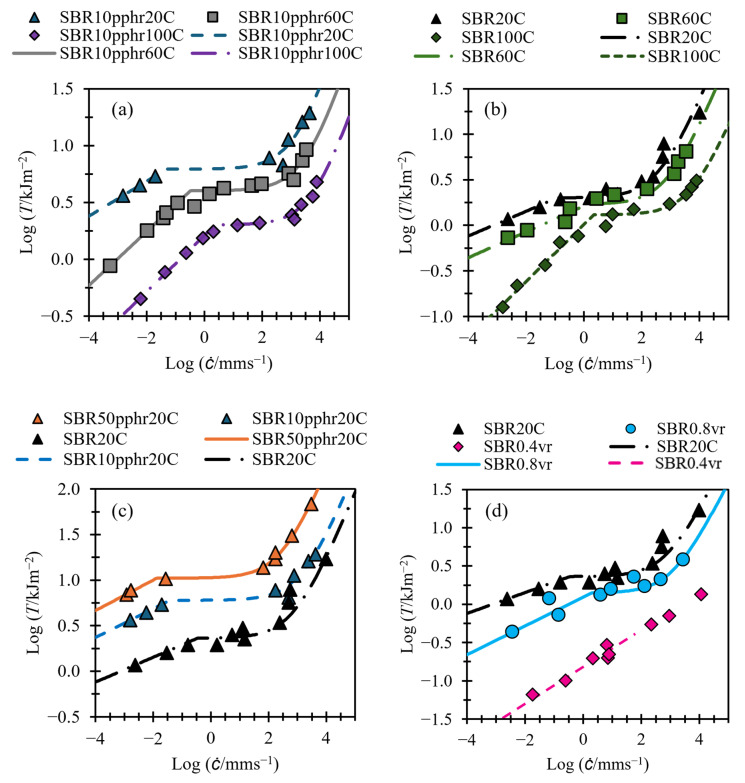
(**a**–**d**) Tearing energy versus crack growth rate for SBR compounds (data taken from Tsunoda [14]). Fits shown with solid and dashed lines. (**a**) SBR filled with N220 carbon black at different temperatures. (**b**) Unfilled SBR at different temperatures. (**c**) SBR with different filler loading of N220 carbon black. (**d**) Unfilled SBR with different levels of swelling.

**Figure 8 polymers-17-01357-f008:**
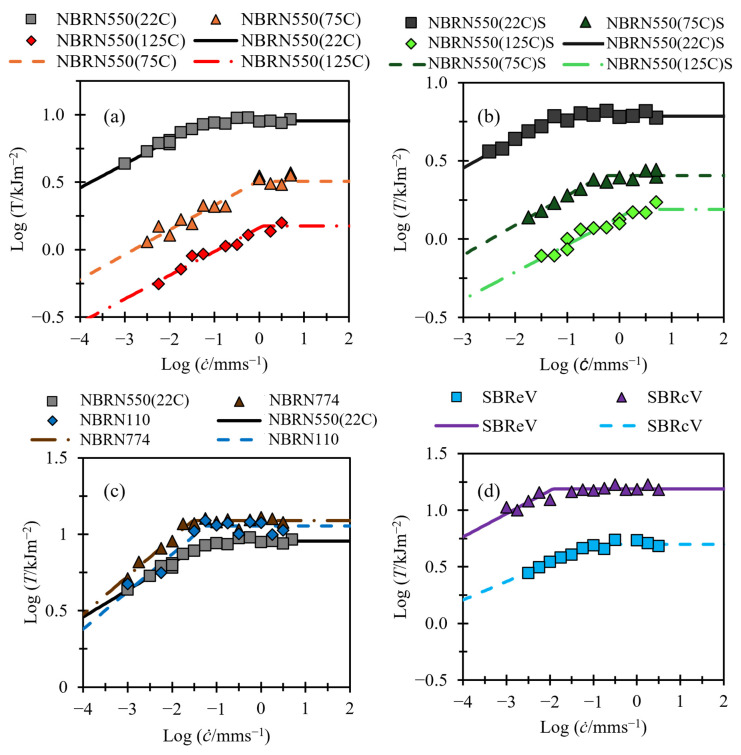
(**a**–**d**) Tearing energy versus crack growth rate. Fits shown with solid and dashed lines. (**a**) NBR filled with N550 carbon black tested at different temperatures. (**b**) NBR filled with N550 carbon black swollen in ULTADRIL for 72 h at 125 °C and tested at different temperatures. (**c**) NBR with different grades of carbon black fillers tested at room temperature. (**d**) SBR with different curing activators tested at room temperature.

**Figure 9 polymers-17-01357-f009:**
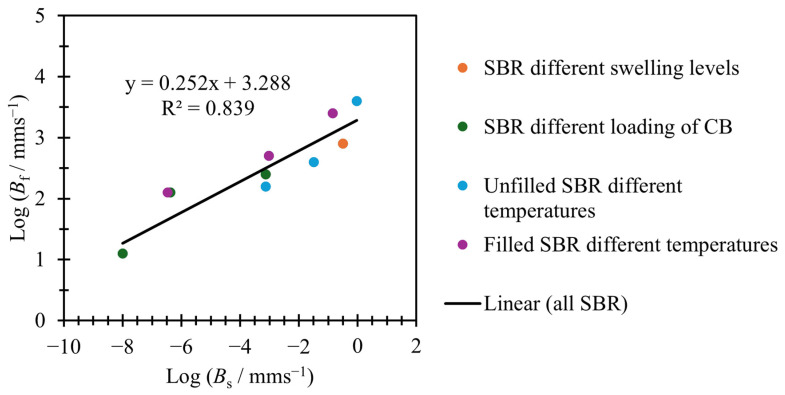
$B_{s}$ versus $B_{f}$ for a range of SBR materials shown in Figure 5.

**Figure 10 polymers-17-01357-f010:**
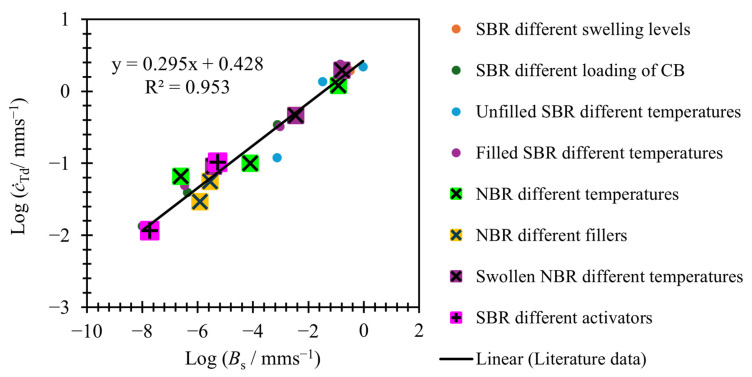
$B_{s}$ versus threshold onset speed for a range of SBR and NBR materials seen in Figure 7 and Figure 8. The linear fit shown in a solid black line is the best fit made from the literature data (shown as solid dots) only.

**Figure 11 polymers-17-01357-f011:**
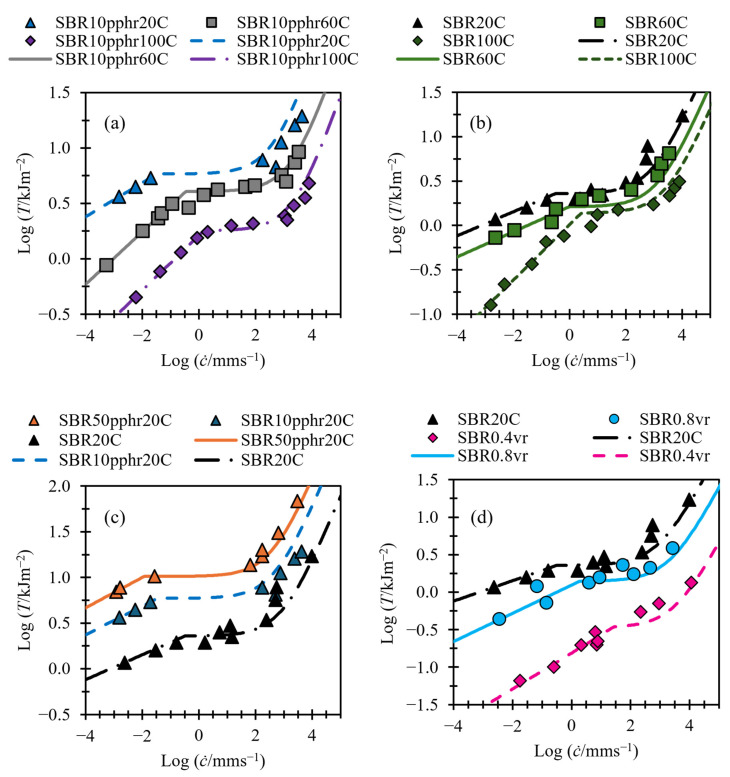
(**a**–**d**) Tearing energy versus crack growth rate for SBR compounds (data taken from Tsunoda [14]). Fits are shown as solid and dashed lines using Equations (8)–(10). (**a**) SBR filled with N220 carbon black at different temperatures. (**b**) Unfilled SBR at different temperatures. (**c**) SBR with different filler loading of N220 carbon black. (**d**) unfilled SBR with different levels of swelling.

**Figure 12 polymers-17-01357-f012:**
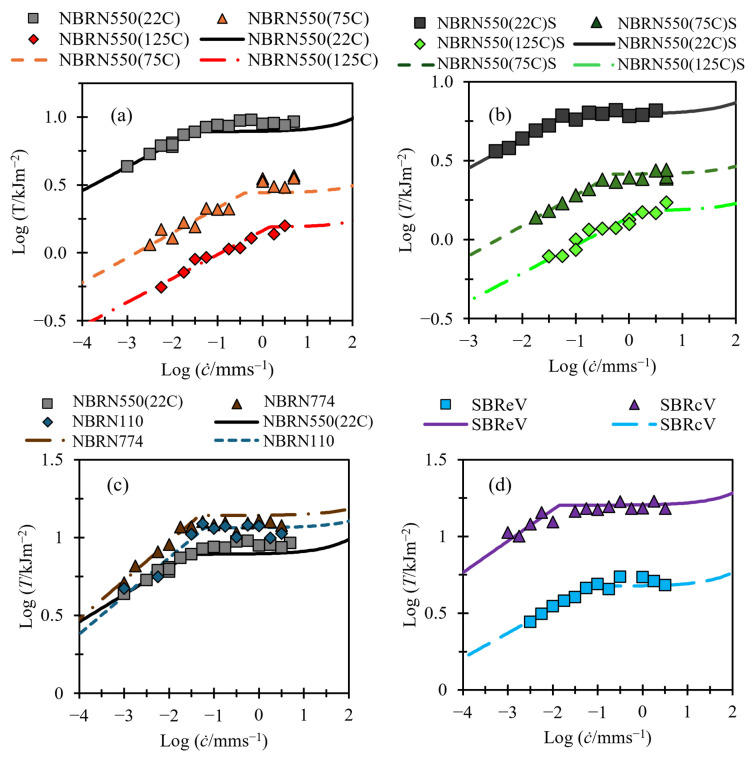
(**a**–**d**) Tearing energy versus crack growth rate. Fits shown with solid and dashed lines using Equations (8)–(10). (**a**) NBR filled with N550 carbon black tested at different temperatures. (**b**) NBR filled with N550 carbon black swollen in ULTADRIL for 72 h at 125 °C and tested at different temperatures. (**c**) NBR with different grades of carbon black fillers tested at room temperature. (**d**) SBR with different curing activators tested at room temperature.

**Table 1 polymers-17-01357-t001:** Material names and details of the base rubber and filler (including parts per hundred rubber (pphr)). Test temperatures and swelling details are also shown.

Material Name	Rubber	Filler	Test Temperature/°C	Notes
SBR20C *	SBR	Unfilled	20	-
SBR60C *	SBR	Unfilled	60	-
SBR100C *	SBR	Unfilled	100	-
SBR10pph20C *	SBR	10 pphr N220	20	-
SBR10pph60C *	SBR	10 pphr N220	60	-
SBR10pph100C *	SBR	10 pphr N220	100	-
SBR50pph20C *	SBR	50 pphr N220	20	-
SBR0.8Vr *	SBR	Unfilled	20	Swollen to 80% rubber by volume in dibutyl adipate
SBR0.4Vr *	SBR	Unfilled	20	Swollen to 40% rubber by volume in dibutyl adipate
NBR_AT ^†^	NBR	Unfilled	20	-
SBR_AT ^†^	SBR	Unfilled	20	-
BR_AT ^†^	BR	Unfilled	20	-
NBRN550(22C)	NBR	60 pphr N550	22	-
NBRN550(75C)	NBR	60 pphr N550	75	-
NBRN550(125C)	NBR	60 pphr N550	125	-
NBRN774	NBR	60 pphr NBRN774	22	-
NBRN110	NBR	60 pphr NBRN110	22	-
NBRN550(22C)S	NBR	60 pphr N550	22	-
NBRN550(75C)S	NBR	60 pphr N550	75	Swollen in ULTADRIL at 125 °C for 72 h
NBRN550(125C)S	NBR	60 pphr N550	125	Swollen in ULTADRIL at 125 °C for 72 h
SBReV	SBR	50 pphr N220	22	Efficient cure
SBRcV	SBR	50 pphr N220	22	Conventional cure

* data taken from Tsunoda, 2001 [14]. ^†^ data taken from Kadir and Thomas, 1981 [13].

**Table 2 polymers-17-01357-t002:** Values of variables used in Figure 6, Figure 7 and Figure 8.

Material Name	$\mathrm{Log}\;(B_{s}$/mms^−1^)	$\beta_{s}$	$T_{d}$/kJm^−2^	$\mathrm{Log}\;(B_{f}$/mms^−1^)	$\beta_{f}$
SBR20C *	−3.13	7.35	6.0	2.20	4/3
SBR60C *	−1.49	7.06	4.1	2.60	4/3
SBR100C *	−0.023	3.20	2.0	3.60	4/3
SBR10pph20C *	−6.47	6.51	2.2	2.05	4/3
SBR10pph60C *	−3.03	4.22	1.9	2.70	4/3
SBR10pph100C *	−0.85	4.05	1.3	3.40	4/3
SBR50pph20C *	−8.00	8.25	10.0	1.10	4/3
SBR0.8Vr *	−0.49	5.34	1.4	2.90	4/3
SBR0.4Vr *	3.42	4.20	-	-	-
NBR_AT ^†^	-	-	4.1	0.9	4/3
SBR_AT ^†^	−2.77	5.10	2.7	2.05	4/3
BR_AT ^†^	−0.48	3.40	>>10	3.40	4/3
NBRN550(22C)	−6.61	5.69	9.0	-	-
NBRN550(75C)	−2.80	5.45	3.1	-	-
NBRN550(125C)	−0.92	5.69	2.1	-	-
NBRN774	−5.92	4.02	12.3	-	-
NBRN110	−5.55	4.08	11.3	-	-
NBRN550(22C)S	−5.43	5.36	6.1	-	-
NBRN550(75C)S	−2.46	5.23	2.55	-	-
NBRN550(125C)S	−0.79	5.70	1.55	-	-
SBReV	−5.27	6.13	5.0	-	-
SBRcV	−7.73	4.88	15.4	-	-

* data taken from Tsunoda, 2001 [14]. ^†^ data taken from Kadir and Thomas, 1981 [13].

## Data Availability

Data are contained within the article.

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
