# Peer review of "A Unified Equation for Predicting Crack Growth in Rubber Composites Across All Crack Growth Rates"

_polymers, 2025, doi:10.3390/polym17101357_

Round 1
Reviewer 1 Report
Comments and Suggestions for Authors
The review comments are attached herewith.

Author Response
We sincerely thank the reviewer for their careful reading of our manuscript and their constructive comments. We greatly appreciate the thoughtful feedback, which has helped us improve the clarity and completeness of the work. Below we address each point in turn.
Comment 1: The novelty of the proposed work should be highlighted in abstract and introduction section.
Response 1: We thank the reviewer for this useful suggestion. We have highlighted the novelty more explicitly in both the Abstract and the Introduction:
- Abstract (page 1, line 24) Added:
“Notably, this is the first model to unify slow, transition, and fast crack growth regimes into a single continuous equation requiring only four variables, enabling prediction of high-speed behavior using only low-speed experimental data — a major advancement over existing six-parameter models.” - Introduction (page 2, line 106, section 1.1, end of section): Added:
“The novelty of this work lies in the development of a single, simplified equation that models the full spectrum of crack growth behaviors — slow, transitional, and fast — using only four variables.”
Comment 2: The model appears to be validated under mode I and quasi-static conditions only. It may not be directly applicable to more realistic loading scenarios such as multiaxial fatigue, impact, or high-frequency cyclic loading, which are common in real-world applications.
Response 2: We thank the reviewer for this insightful comment. To clarify, we have expanded Section 1.3 (page 4, line 139) to highlight that while our experimental data were collected under quasi-static loading conditions, the theoretical framework for tearing energy in rubber applies more broadly. Rivlin and Thomas [7] showed that different crack geometries and loading configurations can be considered equivalent to a local opening (Mode I) fracture at the crack tip, allowing tearing energy to be treated as a material property independent of test mode. We have revised the original sentence:
From:
“Rivlin & Thomas [7] established that these different geometries still return the same tearing energy properties.”
To:
“Rivlin and Thomas further showed that different crack geometries and loading configurations can be considered equivalent to a local opening (Mode I) fracture at the crack tip, allowing tearing energy to be treated as a material property independent of test mode.”
This addition clarifies that although our study focused on Mode I, the model’s formulation aligns with the established equivalence of tearing energy across different geometries and loading modes in rubber.
Comment 3: How does the new model eliminate the need to explicitly define transition points between crack growth regimes?
Response 3: We appreciate this insightful question. To clarify, we have added the following clarification in Section 2.1 (page 7, line 186, after the sentence introducing T_d​):
“While the model retains a threshold energy T_d​ for the slow crack growth term, it eliminates the need to define a separate transition point for fast crack growth (i.e., T_df​ in previous models). Additionally, as will be discussed in Section 3.2, T_d​ can be predicted from slow crack growth data, further reducing the experimental complexity.”
Comment 4: How does fixing the fast crack growth exponent (βf = 4/3) influence model generalizability and accuracy?
Response 4: We thank the reviewer for raising this point. We have addressed it in the Discussion section (page 18, line 406, after the paragraph discussing additional testing needs) by adding:
“The model uses a fixed value of βf=4/3 for the fast crack growth exponent, which provided consistently good fits across all tested rubbers, as shown in Table 2. This choice simplifies fitting by reducing the number of free parameters. However, while the value aligns well with the viscoelastic fracture behavior of non-strain crystallizing rubbers, different material systems with unique fracture mechanisms such as strain-induced crystallization may require re-evaluation of this exponent. Further testing across a broader set of elastomers and fracture modes will help confirm the universality of this fixed exponent.”
Comment 5: What are the limitations of the model, particularly under highly dynamic or multiaxial loading conditions?
Response 5: This comment is closely related to Comment 2, and we addressed both together. As noted in Response 2.
Comment 6: How does the model perform at elevated temperatures, and is there a temperature dependency in the parameters?
Response 6: We appreciate this question. Temperature effects are incorporated in our study, with data collected at multiple temperatures (20°C, 60°C, 100°C, 125°C), as shown in Figures 7 and 8. The temperature-dependent parameters (log(B_s), βs​, and T_d​) are detailed in Table 2. These results demonstrate that the model performs well across a wide temperature range. We note that while parameters are fit separately for each temperature, extending the model to explicitly include temperature dependence is an interesting avenue for future work. No changes were made to the manuscript, as this is already covered.
Comment 7: The study would benefit from a sensitivity analysis or uncertainty quantification to assess the robustness of the proposed model under variation in input parameters like material scatter.
Response 7: We agree that a formal sensitivity analysis would provide valuable additional insight. Although this was not included in the current study, the robustness of the model is demonstrated by successful fitting across a broad range of materials, fillers, and test conditions (Figures 6–8). We have noted this as a valuable direction for future work. No manuscript changes were made in response to this comment.
We trust that these revisions and clarifications fully address the reviewer’s comments and further strengthen the manuscript. Thank you again for your thoughtful and constructive feedback.
Reviewer 2 Report
Comments and Suggestions for Authors
The manuscript "A Simplified Model for Predicting Crack Growth in Rubber Composites Across All Crack Growth Speeds" presents a novel unified equation for modeling the relationship between tearing energy and crack growth speed in elastomers. The approach reduces the number of parameters needed from six to four by treating crack growth as the result of two coexisting energy dissipation mechanisms rather than three distinct regions. While the work has merit and potential significance to the field, several important revisions are needed before publication can be recommended.
Below my detailed comments:
- The manuscript lacks sufficient physical justification for the proposed equation. The authors need to clearly explain the physical basis for the two energy dissipation mechanisms represented in Equation 5.
- The paper essentially presents a new mathematical equation that combines two terms to describe the relationship between tearing energy and crack growth speed, but it doesn't offer a comprehensive physical model that explains the underlying mechanisms. I recommend that the title of the manuscript should better reflect its content with something like: "A Unified Equation for Predicting Crack Growth in Rubber Composites Across All Crack Growth Speeds"
- The threshold energy Td is introduced but its physical significance is not adequately explained. What physical phenomenon does this threshold represent?
- The current validation relies heavily on previously published data. The authors should more clearly differentiate their original experimental contributions from their analysis of literature data.
- The trouser tear test data presented does not extend to high enough crack growth speeds to fully validate the fast crack growth region predictions, which is a significant limitation that must be explicitly acknowledged.
- The manuscript presents intriguing correlations between log(Bs) and both log(Bf) and log(ċTd) but lacks sufficient discussion of why these relationships exist.
- The authors fix βf at 4/3 for all materials without adequate justification. More evidence is needed to support this as a universal value.
- Figure 1 would benefit from additional information on the physical processes that lead to the different crack growth behaviors shown (smooth steady state, rough steady state, stick slip, and knotty tearing). Currently, these are presented primarily as visual observations, but connecting them to underlying material deformation mechanisms would enhance the reader's understanding and strengthen the connection to the proposed mathematical equation.
- The statement that fast crack growth "usually is characterized by a steady smooth crack surface" is an oversimplification that should be qualified.
- Some grammatical errors and typos need correction (e.g.,"significantlty").
- Consistent terminology should be maintained throughout (sometimes "crack growth speed" and sometimes "crack growth rate" are used).
Author Response
We sincerely thank you for your thoughtful and detailed review of our manuscript. We appreciate your insightful comments and constructive suggestions, which have helped us to improve the clarity, rigor, and presentation of the paper.
Below, we address each of your comments point by point, outlining the corresponding changes made to the manuscript or providing clarifications where appropriate.
Comment 1: The manuscript lacks sufficient physical justification for the proposed equation. The authors need to clearly explain the physical basis for the two energy dissipation mechanisms represented in Equation 5.
Response 1: We appreciate this insightful suggestion. In response to this comment, and in alignment with the reviewer’s second comment, we have revised the manuscript title to better reflect the nature of the work as a phenomenological equation rather than a fully mechanistic model. The new title reads:
“A Unified Equation for Predicting Crack Growth in Rubber Composites Across All Crack growth rates.”
The model in this study is formulated as a phenomenological representation of two coexisting energy dissipation processes governing crack growth at different speeds. While we acknowledge the importance of linking these terms to specific micro-mechanisms, the precise underlying mechanisms remain to be fully characterized. This point is already discussed in the Discussion section (page 18, 3rd paragraph), where we note that further work is needed to more rigorously connect the model terms to material-level mechanisms. Therefore, no additional changes were made to the text, as this is an identified area for future research.
Comment 2: The paper essentially presents a new mathematical equation that combines two terms to describe the relationship between tearing energy and crack growth rate, but it doesn't offer a comprehensive physical model that explains the underlying mechanisms. I recommend that the title of the manuscript should better reflect its content with something like: “A Unified Equation for Predicting Crack Growth in Rubber Composites Across All Crack Growth Speeds.”
Response 2: We thank the reviewer for this thoughtful and constructive suggestion. We agree that the recommended title more accurately reflects the scope and nature of the manuscript, emphasizing that the work presents a unified equation rather than a fully mechanistic physical model. Accordingly, we have revised the title of the manuscript to:
“A Unified Equation for Predicting Crack Growth in Rubber Composites Across All Crack Growth Rates.”
We appreciate the reviewer’s recommendation to improve the clarity and precision of the manuscript’s presentation.
Comment 3: The threshold energy Td​ is introduced but its physical significance is not adequately explained. What physical phenomenon does this threshold represent?
Response 3: We thank the reviewer for raising this important point. To clarify the physical interpretation of Td​, we have added the following sentence to Section 2.1 (page 6, line 189):
“The threshold tearing energy Td​ represents the upper limit of the slow crack growth energy dissipation mechanism. The contribution to the total tearing energy from the slow crack growth energy dissipation mechanism cannot increase beyond this energy, even at higher crack growth speeds.”
This addition explains the functional role of Td​ as the limiting tearing energy contribution from the slow mechanism. We acknowledge that while the model defines this threshold phenomenologically, the precise material-level mechanism responsible for this limit remains to be fully characterized and is an area identified for future investigation.
Comment 4: The current validation relies heavily on previously published data. The authors should more clearly differentiate their original experimental contributions from their analysis of literature data.
Response 4: We thank the reviewer for this valuable comment. We agree that it is important to clearly distinguish original experimental results from literature data. We have reviewed the manuscript and ensured that these distinctions are explicitly stated in all relevant sections. To further clarify, we have added the following sentence to Section 2.3 (page 8, line 243):
“The NBR, SBReV, and SBRcV data presented in this study were generated through original trouser tear tests, while additional datasets were digitized from published literature sources as noted in the figure captions and text.”
We believe this addition makes the distinction even clearer for readers and reinforces the transparency of the data sources.
Comment 5: The trouser tear test data presented does not extend to high enough crack growth speeds to fully validate the fast crack growth region predictions, which is a significant limitation that must be explicitly acknowledged.
Response 5: We thank the reviewer for this helpful comment. We agree that the inability of the TT tests to reach sufficiently high crack growth rates represents a limitation in directly validating the fast crack growth region predictions. To clarify this point, we have revised the text above Figure 8 to read (page 12, line 296):
“Figure 8 shows tearing energy data from TT tests. As stated in Section 1.3, TT tests are not able to reach speeds as high as PSCG tests. As a result, the data collected with this method are insufficient to validate the fast crack growth region, which represents a limitation of this study that must be addressed in future work. This data can still be useful for exploring the link between slow tearing energies and the transition energy Td​.”
We believe this revision makes the limitation more explicit and clearer for readers.
Comment 6: The manuscript presents intriguing correlations between log(Bs) and both log(Bf) and log(ċTd) but lacks sufficient discussion of why these relationships exist.
Response 6: We thank the reviewer for highlighting this important point. We agree that these correlations are an interesting and notable finding of the study. While the underlying physical basis for these relationships is not yet fully established, we believe they may reflect interdependencies between the parameters governing energy dissipation at different crack growth rates. We acknowledge that further work is needed to investigate the mechanistic origin of these correlations. We have not added further speculation to the manuscript at this stage to avoid overinterpreting the empirical trends but have identified this as an important direction for future research.
Comment 7: The authors fix βf​ at 4/3 for all materials without adequate justification. More evidence is needed to support this as a universal value.
Response 7: We thank the reviewer for this valuable comment. We agree that clarification of the choice of βf=4/3 was necessary. To address this, we have added clarifying statements in two locations within the manuscript:
- In Section 2.1 (page 6, line 209), we added:
“The value of βf=4/3 was selected empirically, as it consistently provided good fits across all materials studied. While this exponent aligns broadly with expectations for viscoelastic fracture behavior, a detailed theoretical derivation for this specific value remains an open question.” - In the Discussion section (page 18, line 406), we further expanded on this point with:
“The model uses a fixed value of βf=4/3 for the fast crack growth exponent, which provided consistently good fits across all tested rubbers, as shown in Table 2. This choice simplifies fitting by reducing the number of free parameters. However, while the value aligns well with the viscoelastic fracture behavior of non-strain crystallizing rubbers, different material systems with unique fracture mechanisms such as strain-induced crystallization may require re-evaluation of this exponent. Further testing across a broader set of elastomers and fracture modes will help confirm the universality of this fixed exponent.”
These additions clarify that βf​ was selected empirically in the current study, acknowledge its limitations, and outline plans for further investigation across other materials and fracture mechanisms. We would also like to note that the authors are actively working on developing a theoretical explanation as part of ongoing research to be published in future studies.
Comment 8: Figure 1 would benefit from additional information on the physical processes that lead to the different crack growth behaviors shown (smooth steady state, rough steady state, stick slip, and knotty tearing). Currently, these are presented primarily as visual observations, but connecting them to underlying material deformation mechanisms would enhance the reader's understanding and strengthen the connection to the proposed mathematical equation.
Response 8: We thank the reviewer for this insightful suggestion. To better connect the fracture surface morphologies shown in Figure 1 to the underlying physical mechanisms, we revised and expanded the explanation in Section 1.3 (page 5). The updated paragraph describes how each fracture morphology corresponds to specific physical processes—for example, crack path undulations or microvoid formation in rough steady-state tearing; transient toughening mechanisms such as strain-induced crystallization in stick–slip tearing; and stable crack advance with minimal deflection in smooth steady-state tearing. We believe this expanded explanation strengthens the connection between observed fracture morphologies and the underlying energy dissipation mechanisms described by the model.
Comment 9: The statement that fast crack growth “usually is characterized by a steady smooth crack surface” is an oversimplification that should be qualified.
Response 9: We thank the reviewer for this helpful observation. We agree that while fast crack growth is commonly associated with smooth crack surfaces, exceptions can occur depending on material and loading conditions. To address this, we have added the following qualifying sentence after the original statement in Section 1.1 (page 3, line 72):
“However, it is noted that under some material compositions or loading conditions, fast crack growth can exhibit more complex or rough fracture surfaces [3,13,17,18].”
We believe this addition improves the nuance and completeness of the description.
Comment 10: Some grammatical errors and typos need correction (e.g., significantlty").
Response 10: We thank the reviewer for noting these issues. We have carefully proofread the manuscript and corrected typographical errors (including “significantlty” to “significantly”) as well as made minor grammatical improvements throughout to enhance clarity and readability.
Comment 11: Consistent terminology should be maintained throughout (sometimes “crack growth speed” and sometimes “crack growth rate” are used).
Response 11: We thank the reviewer for identifying this inconsistency. We have reviewed the manuscript and standardized the terminology to consistently use “crack growth rate” throughout, ensuring alignment with common terminology in the fracture mechanics literature.
We again thank you for your careful and valuable review. We believe that the revisions and clarifications have strengthened the manuscript, and we hope that the revised version meets your expectations.
Reviewer 3 Report
Comments and Suggestions for Authors
Comment 1) Are the PSCG tests used in the literature equivalent to the TT tests used in this paper in terms of obtaining data? Experimental comparison and elaboration is suggested.
Comment 2) How was βf determined in 2.1? Please add material science theoretical basis or mathematical treatment and process.
Comment 3) It is suggested to add the two energy loss mechanisms in 2.1 that lead to slow and fast crack growth speeds under the perspective of material micro-mechanism.
Comment 4) It is suggested to clarify some more details of the test, such as how many level tests were set up for the same test condition, what is the error between the level test result and the average value, etc.
Author Response
We thank the reviewer for their thoughtful and constructive feedback, which has helped us strengthen and clarify the manuscript. Below we address each of the reviewer’s comments in turn.
Comment 1: Are the PSCG tests used in the literature equivalent to the TT tests used in this paper in terms of obtaining data? Experimental comparison and elaboration is suggested.
Response 1: We thank the reviewer for this valuable comment. The equivalence between PSCG and TT tests in terms of tearing energy measurement is discussed in Section 1.3 (page 5, line 139). As stated there, based on Rivlin & Thomas [7], the tearing energy measured is independent of specimen geometry because energy dissipation occurs almost exclusively near the crack tip. This ensures that both PSCG and TT tests yield the same tearing energy properties when proper experimental techniques are followed. To further clarify, we have added the following statement after Figure 3 (page 5, line 149):
“No direct experimental comparison between PSCG and TT tests was conducted within this study; however, their equivalence in measuring tearing energy is well established in literature [7,16], ensuring the validity of combining data from both methods.”
This addition, supported by both foundational (Rivlin & Thomas [7]) and recent (Sakulkaew et al. [16]) references, elaborates on the equivalence and addresses the reviewer’s request.
Comment 2: How was βf determined in 2.1? Please add material science theoretical basis or mathematical treatment and process.
Response 2: We appreciate the reviewer’s question regarding the determination of βf​. The value of βf=4/3 was selected empirically because it consistently provided good fits across all materials studied. While this exponent aligns with general expectations for viscoelastic fracture behavior, a detailed theoretical basis for this specific value was not established within this study. To clarify this point, we have added the following sentence in Section 2.1 (page 6, line 209):
“The value of βf=4/3 was selected empirically, as it consistently provided good fits across all materials studied. While this exponent aligns broadly with expectations for viscoelastic fracture behavior, a detailed theoretical derivation for this specific value remains an open question.”
This acknowledges the empirical nature of βf​ and outlines plans for future investigation. We would also like to note that the authors are actively working on developing a theoretical explanation as part of ongoing research to be published in future studies.
Comment 3: It is suggested to add the two energy loss mechanisms in 2.1 that lead to slow and fast crack growth speeds under the perspective of material micro-mechanism.
Response 3: We appreciate this insightful suggestion. The model in this study is formulated as a phenomenological representation of two coexisting energy dissipation processes governing crack growth at different speeds. While we acknowledge the importance of linking these terms to specific micro-mechanisms, the precise underlying mechanisms remain to be fully characterized. This point is already discussed in the Discussion section (page 17, 3rd paragraph), where we state that further work is needed to more rigorously connect the model terms to material-level mechanisms. Therefore, no additional changes were made to Section 2.1, as this is an identified area for future research.
Comment 4: It is suggested to clarify some more details of the test, such as how many level tests were set up for the same test condition, what is the error between the level test result and the average value, etc.
Response 4: We thank the reviewer for this helpful suggestion. To provide greater clarity on the experimental methodology and data quality, we have added the following explanation to Section 2.3 (page 8, line 250):
“To maximize the mapping of the crack growth rate versus tearing energy plot, one test was conducted at each crack growth rate at every log(0.25 mm/s) interval, following common practice for this type of study. Samples showing significant early crack growth deviation, for which accurate identification of the propagation energy was difficult, were removed from the data set. For digitized literature data, reproducibility follows that reported by the original authors.”
We believe this addition clarifies the number of tests per condition, the approach taken to balance mapping resolution with experimental efficiency, and the treatment of data quality and reproducibility.
We trust that these clarifications and revisions fully address Reviewer 2’s comments and improve the manuscript. We are grateful for the reviewer’s insightful suggestions, which have helped strengthen the work.
Round 2
Reviewer 1 Report
Comments and Suggestions for Authors
Authors have revised the manuscript as per the suggested comments. Now, the paper may be accepted for possible publication in this journal.
Reviewer 2 Report
Comments and Suggestions for Authors
The revised manuscript addressed all critical issues highlighted in my report. The paper can be accepted in its present form.